# Use of Telemedicine Technology among General Practitioners during COVID-19: A Modified Technology Acceptance Model Study in Poland

**DOI:** 10.3390/ijerph191710937

**Published:** 2022-09-01

**Authors:** Renata Walczak, Magdalena Kludacz-Alessandri, Liliana Hawrysz

**Affiliations:** 1Faculty of Civil Engineering, Mechanics and Petrochemistry, Warsaw University of Technology, 09-400 Plock, Poland; 2College of Economics and Social Sciences, Warsaw University of Technology, 09-400 Plock, Poland; 3Faculty of Management, Wrocław University of Science and Technology, 50-370 Wrocław, Poland

**Keywords:** telemedicine technology acceptance, perceived usefulness, perceived ease of use, primary healthcare

## Abstract

During the COVID-19 pandemic, telehealth became a popular solution for the remote provision of primary care by General Practitioners (GPs) in Poland. This study aimed to assess the GPs’ acceptance of telehealth during the COVID-19 pandemic in Poland and to explain the factors that drive GPs’ need to implement a telehealth system in primary care using the modified Technology Acceptance Model (TAM). In Poland, 361 GPs from a representative sample of 361 clinics drawn from 21,500 outpatient institutions in Poland participated in the empirical study. Structural equation modelling (SEM) was used to evaluate the causal relationships that were formulated in the proposed model. Research has shown that Polish GPs reported a positive perception and high acceptance of the telehealth system during the COVID-19 pandemic. Overall, the results show that the social factors (image, decision autonomy, perception of patient interaction) significantly positively influence the technological factors (perceived ease of use and perceived usefulness) that influence the need to implement a telehealth system. The proposed socio-technological model can serve as a theoretical basis for future research and offer empirical predictions for practitioners and researchers in health departments, governments, and primary care settings.

## 1. Introduction

During the COVID-19 pandemic, many people were afraid to go to healthcare facilities. Supporting patients in performing non-emergency procedures with the use of telemedicine has become a popular solution in such a situation. Telemedicine is a key technology that enables the provision of health services at a distance by healthcare professionals, especially GPs, who play a key role in protecting the health of their patients. Specialized personnel use communication and information technology (ICT) to exchange information in the diagnosis, treatment and prevention of diseases and injuries, for testing and assessment, and for the lifelong learning of health professionals [1,2,3,4].

Telemedicine combines convenience, low cost, and easy access to health-related information and communication using the Internet and related technologies. The use of telemedicine can help patients become more involved in their healthcare plan and increase their autonomy. Telemedicine can significantly improve patients’ health in areas with insufficient access to primary and specialist healthcare. This tool aims to improve access to care for everyone, regardless of location, and reduce face-to-face visits [5,6]. Given the important role of telemedicine in improving healthcare, especially during the COVID-19 epidemic, it may be important to investigate factors that drive the need for GPs to implement this technology. One of the key measures for successful ICT implementation and effective implementation is user acceptance of the technology.

Telemedicine is a valuable health service only if GPs use it proactively; therefore, the GPs’ approach to telemedicine technology plays a key role in its successful use. In addition, GPs act as watchdogs of the telemedicine service by deciding whether it is offered. GPs are the main users and stakeholders of telehealth services, and their acceptance in healthcare facilities profoundly impacts their success. Some GPs may perceive telemedicine technology as a threat to their knowledge and are reluctant to use it, while others may feel a strong need to implement it. When new information systems are implemented in a healthcare organization, GPs must actively participate in the reception of the telemedicine service. Therefore, in order to convince GPs to use telemedicine services, it is important to understand the variables that determine how physicians can change their perception of telemedicine services [7].

Decisions on adopting a given technology are among the most important medical and administrative decisions made in healthcare systems in general and in hospitals in particular [8]. However, the adaptation process is complex and depends on many interacting factors [9,10]. Factors identified in the literature as important for telemedicine adoption have been divided into four levels: environmental, organizational, individual, and innovative. So far, research has focused mainly on the organizational level [11]. Less attention has been paid to the individual level, although decisions about adoption in healthcare facilities are made by (groups of) people and are subject to subjective influence. GPs, in particular, strongly influence the introduction of innovation in healthcare entities as they serve as initiators, facilitators, and decision-makers [12,13,14].

Therefore, this study aimed to analyze GPs’ acceptance of the telehealth system in primary care during the COVID-19 pandemic in Poland and to examine possible factors affecting this acceptance. To investigate these factors, we developed a proprietary telemedicine service acceptance model as a modification of the technology acceptance model (TAM). The basic TAM model assumes the mediating role of technical factors (perceived ease of use and usefulness) in the relationship between the system and user characteristics (external variables) and system acceptance [15]. The model used in this study includes key factors drawn from the original TAM and previous qualitative research on satisfaction from telemedicine services. The data used to develop and validate the scale was obtained from the literature, face-to-face, and telephone interviews with GPs in the pilot study, and a research survey conducted among GPs. The target population of this study were GPs who carried out remote teleconsultations with primary care patients during the COVID-19 pandemic in Poland. In total, 361 respondents took part in the survey. The respondents came from 361 clinics drawn from 21,500 clinics in Poland, one GP from each clinic. The data processing was performed using a method based on the modelling of structural equations in the AMOS 24 application

This article is divided into six sections. Section 2 analyzes the literature on technology acceptance models in healthcare and defines the concepts discussed in the article. The proposed model with its theoretical foundations and research methodology is presented in Section 3. Section 4 presents the research results, and the discussion of these results is included in Section 5. Finally, Section 6 contains the conclusions that end the article.

## 2. Theoretical Basics

Broadly speaking, there are three main types of telemedicine: store-and-forward, remote patient monitoring, and real-time interactive services known as teleconsultations [16]. Store-and-forward telemedicine services are provided asynchronously so that the data exchange process can occur even when the sender and recipient are not present simultaneously [17]. One example is an X-ray of a patient sent to a healthcare professional by email. Real-time interactive services are defined as telemedicine services that require interactive interaction between healthcare professionals and patients at the same time, such as online health consultation services conducted via video [17]. Since the outbreak of the COVID-19 pandemic, also primary healthcare facilities in Poland have started to develop real-time interactive services (telehealth services) to expand the coverage of health services and prevent the spread of the COVID-19 pandemic. It is a very modern form of providing medical services that can be implemented using well-known telecommunications and ICT tools to interact with people who are geographically distant from each other.

This study used the TAM, which is one of the popular and widely used models for studying the social and technological mechanisms of ICT adoption and usage [18,19,20]. Over the past few decades, the TAM model has become the dominant model for explaining technology acceptance by assessing beliefs, attitudes, and intentions towards technology and its actual adoption. Technology acceptance in this model has been defined as the mental state of an individual with regard to his voluntary or deliberate use of a particular technology [21]. The original model treated intention as a direct determinant of behavior, while user attitudes and social norms were predictors of intention. The main goal of TAM was to anticipate the acceptance of information technology and to shed light on design problems related to new information systems prior to their adoption. Most researchers found this model very simple and easy to use, which turned out to be a very powerful model for identifying variables influencing user acceptance of computer technology [7].

From the end-user perspective, this model focuses on the factors determining the behavioral intention of using new ICT [22,23]. According to TAM, the behavioral intention (BI) of an individual to use a system is determined by two beliefs: perceived usefulness (PU) (the degree to which the user believes that using a particular system will improve his performance at work) and perceived ease of use (PEU) (the degree to which a person believes that using a particular system will the system will reduce physical or mental effort). The perceived ease of use also indirectly influences the intention to use through its direct impact on the perceived usefulness [15,21].

Research to date has shown that the main components of TAMs: PU and PEU of ICT, are considered to be the main determinants that directly or indirectly explain BI to use (“accept”) a new technology [24,25,26,27,28]. BI is the degree to which a person has formulated conscious plans to perform or not perform a particular future behavior [29].

A systematic review to examine the factors influencing ICT uptake by healthcare professionals covering all technology acceptance models in health services found that PU and PEU were the two most influential factors in these models [29,30]. PEU is defined as the degree to which a person believes that using the system is easy [15,21]. PEU is considered to be one of the most important TAM constructs that helps predict user acceptance or rejection of technology [31,32]. PU refers to “the extent to which an individual believes that applying a particular technology will improve job performance” [33]. Research on technology acceptance in various fields suggests that PU is the main factor determining the acceptance and application of new technology [34,35,36,37]. On the basis of TAM, several studies have demonstrated the explanatory power of these two factors in interpreting specific behaviors of health professionals, such as adherence to guidelines, the use of health information technology, etc. [38,39,40,41].

In order to explain the acceptance of new technology, the extended and modified models take into account various external variables. Since the introduction of TAM, the initial model has been enhanced with some external constructs to explain user acceptance of new ICT technologies such as information exchange, staff ICT experience, technical infrastructure [42,43], positive social norms, computer skills [44], social influence [45,46], and others [47,48,49,50]. Among the external factors, a special role is played by personal characteristics (e.g., self-efficacy, risk, trust and innovation, experience) [51], social capital factors (social trust, institutional trust, and social participation) [18], social impact (subjective norm, voluntariness and image), cognitive-instrumental processes (adequacy of work, quality of results and the ability to demonstrate results) [36], and organizational features [52]. The most common factors added to the original TAM in almost all technological contexts were, in order of importance and frequency of repetition, compatibility, subjective norm, self-efficacy, experience, training, anxiety, habit, and facilitators [36].

Several studies have also suggested that user attitudes may be important in the acceptance and effectiveness of using technology in practice [53,54,55]. Attitude is a predisposed state of mind regarding the system’s benefits in improving work efficiency, managing work time and its impact on improving the quality of work performed [56]. The attitude variable is usually omitted in some studies due to the argument that it should not be a strong predictor of acceptance, but rather it may be one of many factors determining acceptance [57].

The TAM model has also been tested in the healthcare context and has been proven to be a good model for the predictive BI of GPs to accept telemedicine technology [58,59,60]. The validity of TAM has been tested in various healthcare areas, such as the intention of GPs to use telemedicine technology in Hong Kong [61,62]; patient acceptance of vendor-provided telehealth [63], public health nurses ‘intentions for internet learning [64], mobile computing acceptance factors in healthcare [65], and nurses’ intention to adopt an electronic logistics information system in Taiwan [44,66].

The results of various studies show that the attitude and acceptance of healthcare providers are critical to the successful implementation of the telehealth system in healthcare systems, as they are the system’s primary users [67,68,69]. 

A general overview of the most widely used acceptance models in health services has shown that TAM is the most important model used to identify factors influencing the adoption of information technology in the healthcare system [70]. However, there is still insufficient information on the needs of users of these technologies in terms of adoption and use in primary care, especially during the crisis of the COVID-19 pandemic. This study, therefore, aims to fill this gap by assessing the validity of the modified TAM and identifying the impact of key technological and social factors on healthcare professionals’ needs in terms of adopting telehealth technologies in crisis conditions.

In technology acceptance research, BI is typically used as the dependent variable instead of actual usage and is considered an accurate predictor of future ICT user behavior [63]. In this study, BI has been replaced with a proprietary construct concerning the need to implement a telehealth system (NEC). Until now, researchers have focused more on the socio-economic and technical acceptance factors of telehealth, as well as the relationship between BI regarding telehealth systems and satisfaction with medical services, but did not analyze GPs’ real needs regarding telehealth use [71].

We decided that it was more reasonable to analyze their actual needs than their intentions in a crisis situation, such as the period of the COVID-19 pandemic, which forced GPs to use ICT systems. To the best of our knowledge, there has been no research to date on the need for GPs to implement a telehealth system, taking into account the impact of social and technological factors on this need. Additionally, the analytical methods used in previous studies, such as descriptive analysis, comparison of differences, and linear regression analysis, could not simultaneously determine the relationship between various factors in the mechanism and test the potential mediating effects. Therefore, this study aims to examine GPs’ needs in using the telehealth system (NEC) and its predictors at the social (image (IM), decision autonomy (AUT), perceived interaction with the patient (SIM)), and technological (PU, PEU) factors level using the basic dimensions of TAM and to investigate the entire mechanism using the structural equation model (SEM). The findings will fill the knowledge gap about the factors related to the GPs’ need to use the telehealth system.

## 3. Materials and Methods

### Theoretical Model and Hypotheses

The model for this study is a modification of TAM from studies that analyze user acceptance of telemedicine, especially teleconsultation, in developing countries [72,73]. The theoretical model of this study was also inspired by the integration of TAM with dimensions derived from other models regarding GP satisfaction analysis. 

We kept the technology factors of TAM—PU and PEU. As the research was conducted after the spread of the COVID-19 pandemic in Poland, we removed the dimension of the BI to use teleconsultation. We added the dimension of the actual needs (NEC) of GPs to implement teleconsultation. 

To comprehensively investigate the mechanism shaping the GPs’ needs to implement a telehealth system, we took into account the social factors that include variables that are directly related to the GPs as the system user: IM, AUT, SIM. The developed research model consists of 5 factors regarding exogenous variables (IM, AUT, SIM, PU, PEU), two factors as endogenous variables (PU, NEC) and 26 indicators. The proposed theoretical model is presented in Figure 1.

The model is based on previous research that suggested that user acceptance depends on two key factors: PU and PEU. In addition, TAM includes three independent variables: IM, AUT, SIM.

In this study, we hypothesized that the constructs and associations described in the modified TAM are appropriate for measuring the NEC by GPs in crisis conditions. In the model shown in Figure 1, the attitude variable that mediates some of the effects of PU and PEU has been removed. 

PU and PEU in the initial TAM were the most dominant determinants of the use of technology [21] and the acceptance of telemedicine services [50,74]. PU was originally defined as the degree to which an individual believed using the system would improve their performance. In this study, the definition of PU was adapted to the specificity of healthcare, where professional utility and performance have a slightly different meaning than those in the original TAM. We assume that GPs believe telehealth is useful when it improves patient care, leads to faster healthcare delivery, better documentation, shorter delivery times, and allows for accurate, low-cost medical monitoring [75,76]. Generally, a PEU is defined as the degree to which a person believes that using a particular system will be effortless [21]. In this study, PEU is to determine whether the telehealth service is easy to learn and use by GPs [77,78].

The variables for examining the NEC were developed on the basis of previous studies on medical staff satisfaction with telephone-based telemedicine during COVID-19 pandemic [79]. The NEC dimension used in this study also served to see if telemedicine is needed in emergencies such as COVID-19 and regardless of emergencies such as COVID-19 and whether telemedicine can replace partially in-person visits.

Both PU and PEU were conceived as key factors in the acceptance of the new technology [75,76,80,81]. Earlier studies in other areas have emphasized the influence of PU and PEU on BI to use [49,50,63,74,82]. In this study, we intend to see if GPs will feel a greater need for a telemedicine service when they find it easy to use, effective, and that it delivers good healthcare outcomes. Therefore, hypotheses 1a, 1b, and 1c were proposed:

**Hypothesis** **1a** **(H1a).***The need for GPs to implement a telehealth system is influenced by the PU of the system*.

**Hypothesis** **1b** **(H1b).***The need for GPs to implement a telehealth system depends on the PEU of the system*.

**Hypothesis** **1c** **(H1c).***The PU mediates the positive relationship between PEU and**NEC*.

Moreover, previous studies have shown that PEU directly influences PU [50,74,83,84]. In other words, the greater ease of use of telehealth suggests that it is more useful for users. In this context, we can conclude that if the GP experienced increased PEU, the technology would probably have been better viewed in terms of its usefulness. Therefore, we proposed that greater PEU inevitably leads to greater PU of telemedicine services.

**Hypothesis** **2** **(H2).***The PEU will have a positive effect on the PU*.

On our scale, we relied on the literature concerning not only the acceptance but also the motivation of GPs, and we also took into account the orientation of GPs towards patients. In healthcare, GPs take care of the decisions about adopting technologies to treat patients. Therefore, it is necessary to incorporate additional factors that can capture GPs’ motivation to improve patient benefits through the use of medical technologies [85,86]. In this study, we considered external social factors such as IM, AUT, and SIM.

The impact of IM and AUT (defined as “voluntariness”) on the acceptance of the telehealth system has already been carried out in Australian long-term care facilities. These studies assumed that these factors could also be important determinants of PU and PEU of a health information system [44]. In other research on GPs’ AUT and GPs-patient interaction, these factors were included in the organizational contextual factors. These factors were also identified on the basis of a literature review and previous empirical research [47,87,88,89].

The IM variable has already appeared in the modified version of TAM2 [49] among the “external variables” included in the social impact (subjective norm, voluntary, and image). The literature defines the IM as the degree to which a person perceives that applying innovation improves their status in their social system [85]. IM can positively affect PU and the NEC by increasing the strength and influence of elevated status, as individuals often respond to social normative influences to establish or maintain a favorable image in the reference group [49].

The AUT and the SIM are included in TAM extensions, which introduce a wide range of external factors [89,90]. Furthermore, the AUT affects the differences between GPs and other user groups in terms of adopting new information technologies. For this reason, GPs are very sensitive to upcoming work environment changes [90]. AUT has been included as an external voluntary variable and defined as “the extent to which potential adopters perceive the decision to adopt a system as optional” [91].

SIM serves as a means of educating patients about their healthcare, including health status assessment and disease diagnosis. Elements of effective SIM interaction during teleconsultation include setting the appropriate tone, accurate interpretation of communication signals, and active listening. Effective SIM is essential to ensure high-quality patient care [92]. This study’s perception of SIM concerns the possibility of understanding the patient, the quality of communication with him, and the possibility of providing reliable advice using the telehealth system. It has already been found in the studies that the relational effect, which relates to the GPs’ perception of interaction with the patient, is a key factor that may shape the NEC and influence its implementation [93]. Several studies have also shown that SIM is associated with intentions to use health-related information technology [94]. On the basis of previous research, we suspect that the perception of interactions with the patient directly impacts the acceptance of medical technology by GP.

The corresponding hypotheses (H) are as follows:

**Hypothesis** **3a** **(H3a).***The need for GPs to implement a telehealth system is influenced by the IM*.

**Hypothesis** **3b** **(H3b).***The need for GPs to implement a telehealth system is influenced by their AUT*.

**Hypothesis** **3c** **(H3c).***The need for GPs to implement a telehealth system is influenced by SIM*.

Items measuring PU and PEU were taken from previously validated questionnaires and modified to fit telehealth in primary healthcare in Poland. Some statements have been developed specifically for this study. The respondents agreed or disagreed with the statements using a five-point Likert-type scale. Selected users and experts conducted preliminary and pilot studies, after which the statements were modified to be appropriate to the context of using telehealth in primary care in Poland. Table 1 lists the statements used in the questionnaire. Table 2 shows the descriptive statistics of variable used in the model.

The survey was carried out by an external company at the request of the Warsaw University of Technology. The survey was positively evaluated by the Professional Ethics Committee of the Warsaw University of Technology.

The data collected were complete, with no missing responses. GPs of primary healthcare institutions filled in a total of 361 questionnaires. A Likert scale was used to evaluate the statements: 1—do not agree, 2—I do not agree some-what, 3—Neither agree nor disagree, 4—I agree somewhat, 5—I agree.

## 4. Results

### 4.1. Exploratory Factor Analysis

A factor analysis was conducted using Principal Component Analysis to identify latent variables. Varimax rotation was used. The variables presented in Table 1 were used for the analysis. The model turned out to be adequate since all variables are sufficiently correlated and form a reliable solution. The adequacy was confirmed by two tests: (1) the Kaiser–Meyer–Olkin test (KMO coefficient) (KMO = 0.90 > 0.8) and the Bartlett’s significant sphericity test (χ^2^ = 5588.27, df = 325, *p* < 0.0001 < 0.05) and (2) extracted communalities. The principal components method and Varimax rotation were used to identify significant factors. Extracted communalities, ideally, should be greater than 0.5. However, the 0.3 threshold is accepted. All variables in the model extracted communalities greater than 0.3 (Table 3). The factor analysis solutions error was small, the number of non-redundant residuals equaled 1.191 × 10^−7^ < 0.05. This allowed all variables to be taken into account in EFA analysis [101].

The model explains almost 70% of variance (Table 4). The results of the exploratory factor analysis are presented in Table 5, where only correlation coefficients greater than 0.5 were included. Six latent variables were expected to be extracted, and six factors were extracted based on eigenvalues greater than 1.

Convergent validity of the model was confirmed. All variables loadings were greater than 0.5, and the average loading per latent factor was greater than 0.7 (Table 5), that means that each latent factor variable were highly correlated. None of initial variables were removed. Additionally, discriminant validity was confirmed. All latent factors are unique, discriminant from each other. Variables load clearly on latent factors. There were no cross-loadings. The reliability of the model was also confirmed. Cronbach’s Alpha coefficient for each latent factor was greater than 0.6, which is permissible [102]

### 4.2. Confirmatory Factor Analysis

On the basis of the EFA model the Confirmatory Factor Analysis model was prepared (Figure 2). $\chi^{2}\left(cmin\right)=739.104;$ DF=284; *p*-value < 0.0001; cminDF=2.602<3; The model fit values: CFI=0.916>0.9; RMSEA=0.067<0.1; $SRMR=0.0598\langle 0.8\;\rangle 0.9$ allowed it to be accepted for further analysis. The strengths of the variables’ correlations described the Average Variance Extracted (AVE) coefficient and composite reliability coefficient (CR). On the basis of those coefficients, the convergent validity of the model could be confirmed. AVE values should be greater than 0.5, CR values should be greater than 0.7. For NEC and PEU variables convergent validity was confirmed only on the basis of the CR coefficient (Table 6). On the basis of the CR factor, reliability was also confirmed [103]. Discriminant validity was confirmed on the basis of the hetero-trait–mono-trait coefficients (HTMT) (Table 7). Coefficient values should be smaller than 0.9 [103]. On this basis, it was possible to create a structural model.

### 4.3. Structural Model

Figure 3 shows a structural model of acceptance of remote medical advice technology. Hypothesis H3a regarding the influence of image on the need for remote advice was not confirmed. This means that GPs did not take up remote work because of their image in society and their reputation among other GPs. Image assessment was not important in the decision to provide remote consultations.

Hypothesis H3c on the effect of the perception of interaction with the patient on the need for teleconsultations was also not confirmed. GPs rated the similarity of the provision of remote consultations to in-patient consultations at a level of 4 on a scale of 1 to 4. It is important to recognise that the ratings for patient contact, both remote and face-to-face, are similar. The variable SIM had no impact on GPs’ evaluation of remote working.

The effect of the decision-making autonomy on the need to give teleconsultations (hypothesis H3b) was small. The AUT variable concerned the voluntariness of remote advice. GPs ranked highly the autonomy to decide whether to provide remote advice. Each variable scored above 4 on a scale of 1 to 5. This means that GPs were not forced by their employer to work remotely. The AUT variable had little impact on GPs’ evaluation of remote consultations (NEC).

The hypothesis H1c regarding an indirect effect of the ease of use of the system on the perceived usefulness on the need for teleconsulting was confirmed. The effect of the ease of use of the system on the need to give teleconsultations (hypothesis H1b) was small and negative. This means that even a system that was difficult to use, this did not cause reluctance to give teleconsultations. The largest regression coefficients in the model were the effect of the ease of use of the system on the perceived usefulness of the system (hypothesis H2) and the perceived usefulness on the need to use the system (hypothesis H1a). The regression coefficients of the structural model and the probability value for the coefficients are summarised in Table 8.

The data collected in the survey were analyzed using exploratory factor analysis (EFA), which opted for a six-factor solution to describe the GPs’ need to implement the system adequately. Then, a confirmatory factor analysis (CFA) was performed, which confirmed the EFA results. Our tests have shown good reliability and validity of the measuring scale.

## 5. Discussion

We attempted to identify predictive constructs influencing the decision of GPs to adopt telemedicine services. The analysis of these constructs from the individual level is important because of GPs’ strong influence on the introduction of innovation in healthcare entities. GPs serve as initiators, facilitators, and decision-makers in those entities. For this purpose, we developed the socio-technological model by modifying the original TAM. We also tested the applicability of the socio-technological model in Poland. In technology acceptance models, behavioral intentions were usually used as the dependent variable considered an accurate predictor of future ICT user behavior [63]. In this study, behavioral intentions were replaced with a proprietary construct concerning the need to implement a telehealth system. We decided that it was more reasonable to analyze GPs’ actual need to implement a telehealth system than their intentions in a crisis situation. During the COVID-19 pandemic, many people were afraid to go to healthcare facilities. The COVID-19 pandemic showed that telehealth is needed in new, emerging and crisis situations and can partially replace in-person patient visits. By being able to use telehealth, patients have easier access to healthcare. Sometimes it is the only way of access to healthcare.

To the best of our knowledge, there has been no research to date on the need for GPs to implement a telehealth system, taking into account the impact of social and technological factors on this need. This study, therefore, fills this gap by assessing the validity of the modified TAM and identifying the impact of key technological and social factors on GPs’ needs in terms of adopting telehealth technologies in crisis conditions. Our results show that perceived usefulness and perceived ease of use significantly impacted the need to implement a telehealth system. Research examining the factors influencing ICT uptake by GPs covering all technology acceptance models in health services found that perceived usefulness and perceived ease of use were the two most influential factors in these models [29,30]. Several studies have demonstrated the explanatory power of perceived usefulness and perceived ease of use in interpreting the specific behaviors of GPs, such as adherence to guidelines, the use of health information technology, etc. [38,39,40,41]. The results of the analysis of the use of telehealth services showed that the main factors influencing the use of telemedicine services, in addition to perceived usefulness and ease of use, were the following variables: social impact, favorable conditions, and trust [73]. Our results showed that the GPs’ image assessment in society was not important in the decision to provide remote consultations. GPs ranked highly the autonomy to decide whether to provide remote advice. Our results show that decision autonomy impacted the need to implement a telehealth system. Previous studies have shown that perceived ease of use directly influences perceived usefulness [50,74,83,84]. This means that the greater perceived ease of use of telehealth suggests that it is more useful for users. GPs who experienced increased perceived ease of use technology would probably better view its usefulness.

Our results show that perceived usefulness mediated the positive relationship between perceived ease of use and the need for GPs to implement a telehealth system.

### 5.1. Contribution

This is the first reliable and accurate scale to measure the physicians’ need to implement a telehealth system in an emergency. As far as we know, our model is the first that has been specifically designed to explain the influence of individual factors at the GP level on the acceptance of a technology system in a healthcare context.

### 5.2. Limitations

However, these studies did not review psychological parameters, such as habits or cultural factors of patients and GPs that could explain the adoption of telemedicine services in developing countries.

## 6. Conclusions

Despite the spread of telemedicine technologies, the acceptance of telehealth services in real healthcare conditions is slow. In this research, we developed a theoretical socio-technological model explaining the predictive factors influencing the need of GPs to use telehealth technologies in the provision of health services. The model is based on the modified technology acceptance model (TAM), taking into account the predictive technological constructs from previously published telemedicine literature (perceived usefulness and perceived ease of use) and three external social factors (1) decision-making autonomy, (2) image, (3) perceived perception of interaction with the patient. 

The study showed that the acceptance model of telehealth services is feasible and can explain GPs’ acceptance of telemedicine services. These results allow us to identify important factors increasing the involvement of GPs in telehealth practice.

Our results confirm the validity of the original TAM constructs. The results show that perceived ease of use and perceived usefulness significantly impact the need to implement a telehealth system. The results also show that decision autonomy impact the need to implement a telehealth system. Moreover, the perceived usefulness of the telehealth system depends on the perceived ease of use and impact on the need to implement a telehealth system.

The proposed model can serve as a basis for future research and offer empirical predictions for practitioners and researchers in health departments, governments, and primary care settings.

This research can help primary healthcare facilities and the government in Poland to provide guidance in developing telemedicine applications. The results can be used as a benchmark for primary healthcare management to analyze the advantages and disadvantages of their telehealth services compared to similar services.

Identifying the most important factors influencing the acceptance of telemedicine from the perspective of GPs, as key players in telehealth projects, can help managers and decision-makers make the right decisions about the successful implementation of telehealth services, especially in the initial stages. Planners and managers should ensure that the telehealth system deployed in the primary care facility is useful and easy to use. GP support by extending their decision-making autonomy can be important to success. Our findings may be also useful in other emergencies or strengthen routine healthcare. Telemedicine may be particularly supportive in cases of routine health issues and when there has been an already established relationship with a GP. The potential of telemedicine can be used wherever it is possible to improve medical processes, the quality of services, and patient safety or achieve a sustainable economy. Future research may also evaluate the aggregation of the factors identified in this article.

## Figures and Tables

**Figure 1 ijerph-19-10937-f001:**
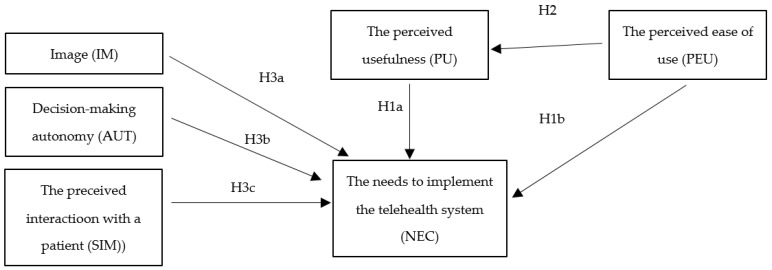
The theoretical model.

**Figure 2 ijerph-19-10937-f002:**
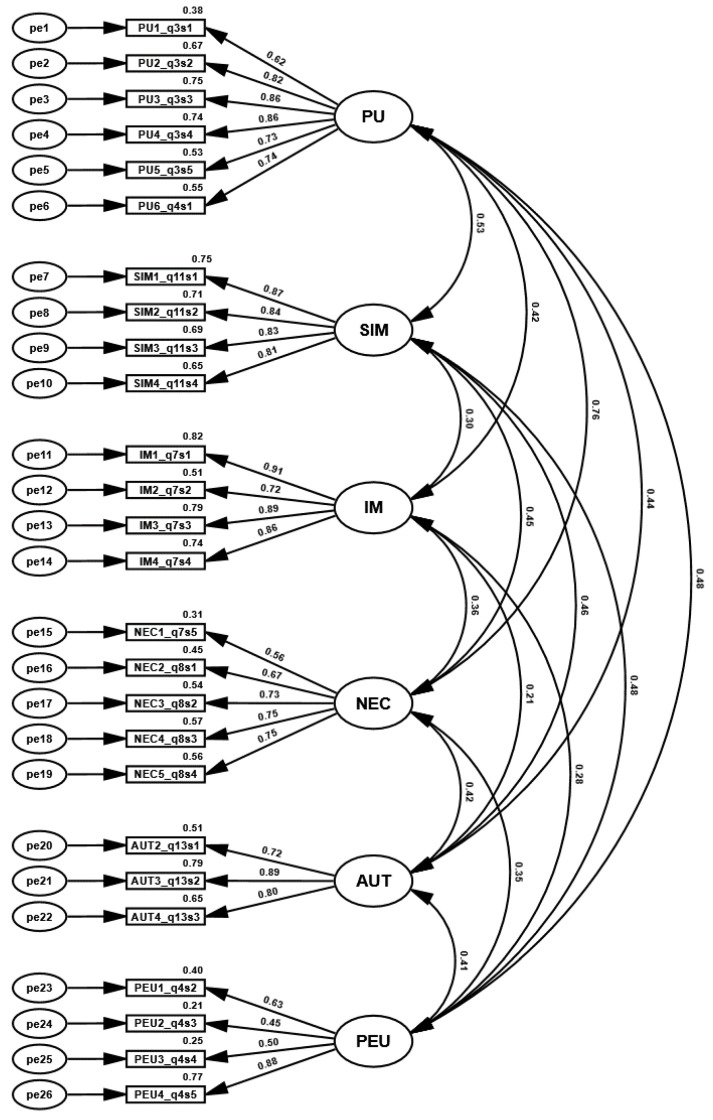
Confirmatory Factor Analysis model. Source: Authors’ own research.

**Figure 3 ijerph-19-10937-f003:**
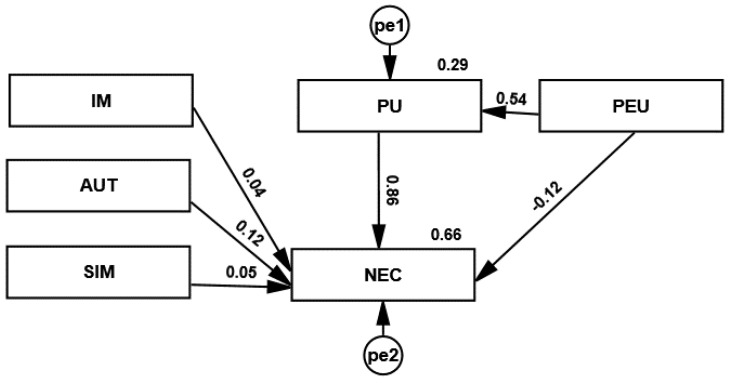
Structural model of the acceptance of remote medical advice technology. Source: Authors’ own research.

**Table 1 ijerph-19-10937-t001:** Statements used in the questionnaire *.

Latent Factor	Variable Name	Statement in the Questionnaire	Literature Source
Perceived Usefulness	PU1_q3s1	My work during a pandemic would be difficult without teleconsultations	Martínez et al., 2006 [80] Rimmer et al., 2020 [95] Bakken et al, 2006 [96] Rho et al., 2014 [7] Davis, 1989 [21]
	PU2_q3s2	Teleconsultations meet my needs at work	
	PU3_q3s3	Teleconsultations increase the efficiency of my work	
	PU4_q3s4	In general I find the teleconsultations a useful system in my work	
	PU5_q3s5	teleconsultations save my time	
	PU6_q4s1	Teleconsultations makes my work easier	
Perceived Ease of Use	PEU1_q4s2	Using a teleconsultations system is easy	Whitten et al., 2005 [97] Martínez et al., 2006 [80] Rimmer et al., 2020 [95] Bakken et al, 2006 [96] Zaidi et al., 2008 [77] Taylor, 2005 [78] Rho et al., 2014 [7] Davis, 1989 [21]
	PEU2_q4s3	Using a teleconsultations system does not require too much intellectual effort	
	PEU3_q4s4	Using the teleconsultations system is understandable for me	
	PEU4_q4s5	Using the teleconsultations system I can do everything I want	
Image	IM1_q7s1	People who use teleconsultations are more prestigious than those who do not use it	Holden, Karsh, 2010 [98] Chau, Hu, 2002 [60] Chismar, Wiley-Patton, 2002 [58]
	IM2_q7s2	People who use teleconsultations get noticed	
	IM3_q7s3	Using teleconsultations is a status symbol	
	IM4_q7s4	I compare myself with people who use teleconsultations	
Needs to implement the telehealth system	NEC1_q7s5	Teleconsultations is an acceptable method of delivering health services	Rho et. al., 2014 [7] Holden, Karsh, 2010 [98] Adewale, 2015 [99]
	NEC2_q8s1	Teleconsultations are needed in new situations, such as the COVID-19 pandemic	
	NEC3_q8s2	Teleconsultations are needed regardless of emerging situations, such as COVID-19	
	NEC4_q8s3	Teleconsultations can partially replace in-person patient visits	
	NEC5_q8s4	By being able to use teleconsultations, patients have easier access to healthcare	
Decision autonomy	AUT2_q13s1	I can influence the number of teleconsultations I take per day	Holden, Karsh, 2010 [98] Yi et.al., 2006 [100] Chau, Hu, 2002 [60]
	AUT3_q13s2	I can decide in which situation to use the teleconsultation	
	AUT4_q13s3	I can decide how the teleconsultations will be done	
Perception of Interaction with the Patient	SIM1_q11s1	When talking to the patient, I understand what the patient’s problem is	Rho et al., 2014 [7] Holden, Karsh, 2010 [98]
	SIM2_q11s2	In conversation with the patient I can easily give advice	
	SIM3_q11s3	I can easily talk to the patient during the teleconsultation	
	SIM4_q11s4	I can understand the patient’s problem	

* Source: Authors’ own research.

**Table 2 ijerph-19-10937-t002:** Descriptive statistics of variable used in the model *.

Variable Name	Mean	Std. Deviation	Variance	Skewness	Kurtosis
PU1_q3s1	4.36	0.935	0.875	−1.731	2.773
PU2_q3s2	4.04	1.031	1.062	−1.247	1.071
PU3_q3s3	3.85	1.176	1.383	−0.912	−0.136
PU4_q3s4	4.24	0.867	0.752	−1.342	1.933
PU5_q3s5	3.89	1.138	1.295	−0.810	−0.378
PU6_q4s1	4.03	1.006	1.013	−1.163	0.990
IM1_q7s1	2.70	1.318	1.738	0.221	−1.077
IM2_q7s2	3.09	1.243	1.544	−0.148	−0.945
IM3_q7s3	2.64	1.331	1.771	0.304	−1.062
IM4_q7s4	2.57	1.367	1.868	0.310	−1.184
SIM1_q11s1	3.98	0.925	0.855	−1.143	1.017
SIM2_q11s2	3.91	1.011	1.022	−0.983	0.293
SIM3_q11s3	3.81	1.050	1.103	−0.731	−0.434
SIM4_q11s4	3.93	0.992	0.984	−0.954	0.223
PEU1_q4s2	4.28	0.834	0.695	−1.338	1.885
PEU2_q4s3	3.81	1.420	2.016	−0.988	−0.467
PEU3_q4s4	4.51	0.671	0.451	−1.819	5.271
PEU4_q4s5	4.06	1.039	1.080	−1.273	1.138
AUT2_q13s1	4.05	1.009	1.017	−1.189	0.907
AUT3_q13s2	4.29	0.818	0.669	−1.660	3.857
AUT4_q13s3	4.35	0.756	0.572	−1.646	4.272
NEC1_q7s5	4.05	0.866	0.750	−1.134	1.725
NEC2_q8s1	4.59	0.631	0.399	−1.864	5.056
NEC3_q8s2	4.30	0.853	0.727	−1.751	3.985
NEC4_q8s3	4.11	0.925	0.857	−1.454	2.321
NEC5_q8s4	4.11	0.872	0.760	−1.279	2.063

* Source: Authors’ own research.

**Table 3 ijerph-19-10937-t003:** Extracted communalities during Exploratory Factor Analysis *.

Variable	Initial	Extraction
PU1_q3s1	1.000	0.546
PU2_q3s2	1.000	0.721
PU3_q3s3	1.000	0.809
PU4_q3s4	1.000	0.789
PU5_q3s5	1.000	0.707
PU6_q4s1	1.000	0.638
PEU1_q4s2	1.000	0.702
PEU2_q4s3	1.000	0.604
PEU3_q4s4	1.000	0.440
PEU4_q4s5	1.000	0.664
NEC1_q7s5	1.000	0.469
NEC2_q8s1	1.000	0.643
NEC3_q8s2	1.000	0.647
NEC4_q8s3	1.000	0.674
NEC5_q8s4	1.000	0.644
IM1_q7s1	1.000	0.835
IM2_q7s2	1.000	0.671
IM3_q7s3	1.000	0.838
IM4_q7s4	1.000	0.817
AUT2_q13s1	1.000	0.755
AUT3_q13s2	1.000	0.810
AUT4_q13s3	1.000	0.760
SIM1_q11s1	1.000	0.774
SIM2_q11s2	1.000	0.757
SIM3_q11s3	1.000	0.728
SIM4_q11s4	1.000	0.755

* Extraction Method: Principal Component Analysis. Source: Authors’ own research.

**Table 4 ijerph-19-10937-t004:** Total variance explained by six factor extracted during Exploratory Factor Analysis *.

Component	Initial Eigenvalues			Extraction Sums of Squared Loadings			Rotation Sums of Squared Loadings		
	Total	% of Variance	% Cumulative	Total	% of Variance	% Cumulative	Total	% of Variance	% Cumulative
1	8.887	34.181	34.181	8.887	34.181	34.181	3.707	14.257	14.257
2	2.506	9.639	43.820	2.506	9.639	43.820	3.451	13.272	27.529
3	2.224	8.555	52.374	2.224	8.555	52.374	3.297	12.681	40.210
4	1.871	7.196	59.570	1.871	7.196	59.570	3.099	11.917	52.127
5	1.583	6.088	65.658	1.583	6.088	65.658	2.323	8.933	61.060
6	1.125	4.327	69.985	1.125	4.327	69.985	2.320	8.925	69.985
7	0.804	3.092	73.078						

* Extraction Method: Principal Component Analysis. Source: Authors’ own research.

**Table 5 ijerph-19-10937-t005:** Rotated Component Matrix *.

Variable	Component					
	1. PU	2. SIM	3. IM	4. NEC	5. AUT	6. PEU
Cronbach’s Alpha	0.894	0.902	0.908	0.815	0.83	0.676
PU1_q3s1	0.623					
PU2_q3s2	0.720					
PU3_q3s3	0.811					
PU4_q3s4	0.733					
PU5_q3s5	0.752					
PU6_q4s1	0.628					
PEU1_q4s2						0.807
PEU2_q4s3						0.670
PEU3_q4s4						0.592
PEU4_q4s5						0.696
NEC1_q7s5				0.545		
NEC2_q8s1				0.760		
NEC3_q8s2				0.727		
NEC4_q8s3				0.765		
NEC5_q8s4				0.659		
IM1_q7s1			0.874			
IM2_q7s2			0.757			
IM3_q7s3			0.897			
IM4_q7s4			0.879			
AUT2_q13s1					0.811	
AUT3_q13s2					0.848	
AUT4_q13s3					0.781	
SIM1_q11s1		0.827				
SIM2_q11s2		0.800				
SIM3_q11s3		0.775				
SIM4_q11s4		0.822				

* Extraction Method: Principal Component Analysis, Rotation Method: Varimax with Kaiser Normalization, Rotation converged in 6 iterations. Source: Authors’ own research.

**Table 6 ijerph-19-10937-t006:** Convergent validity measures *.

Variable	CR	AVE	MSV	$\sqrt{AVE}$	PU	SIM	IM	NEC	AUT	PEU
					Correlations					
PU	0.899	0.602	0.575	0.776	1					
SIM	0.904	0.701	0.277	0.837	0.526	1				
IM	0.909	0.717	0.180	0.847	0.424	0.304	1			
NEC	0.823	0.484	0.575	0.696	0.758	0.454	0.363	1		
AUT	0.847	0.650	0.209	0.806	0.436	0.457	0.206	0.417	1	
PEU	0.719	0.407	0.233	0.638	0.483	0.479	0.280	0.347	0.411	1

* *p*-value < 0.0001. Source: Authors’ own research.

**Table 7 ijerph-19-10937-t007:** HTMT analysis *.

	PU	SIM	IM	NEC	AUT	PEU
PU						
SIM	0.551					
IM	0.444	0.335				
NEC	0.762	0.468	0.383			
AUT	0.450	0.455	0.241	0.448		
PEU	0.466	0.419	0.290	0.323	0.440	

* *p*-value < 0.0001. Source: Authors’ own research.

**Table 8 ijerph-19-10937-t008:** Standardized regression weights for the structural model.

Hypothesis	Variables	Relation	Variables	Standardized Path Estimate	*p*-Value	Confirmation of the Hypotheses
H2	PU	←	PEU	0.543	<0.0001	Supported
H3a	NEC	←	IM	0.036	0.268	Not supported
H3b	NEC	←	AUT	0.111	0.002	Supported
H1a	NEC	←	PU	0.791	<0.0001	Supported
H1b	NEC	←	PEU	−0.108	0.004	Supported
H3c	NEC	←	SIM	0.049	0.205	Not supported
H1c	PEU	→ PU →	NEC	0.464	0.001	Supported

## Data Availability

Not applicable.

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
