# Peer review of "Use of Telemedicine Technology among General Practitioners during COVID-19: A Modified Technology Acceptance Model Study in Poland"

_ijerph, 2022, doi:10.3390/ijerph191710937_

Round 1

Reviewer 1 Report

The manuscript “Acceptance of telemedicine technology among General Practitioners during Covid-19 in Poland” is well written. Utilizing telehealth enhances the delivery of health care. Consequently, telemedicine should be an integral component of health care during COVID-19, ensuring the safety of patients and healthcare professionals. I have minor suggestions.

Add study design in the title

Use of telemedicine technology among General Practitioners during Covid-19: A modified Technology Acceptance Model study in Poland

The author may strengthen the discussion section.

Add following SR

Monaghesh, E., & Hajizadeh, A. (2020). The role of telehealth during COVID-19 outbreak: a systematic review based on current evidence. BMC public health20(1), 1-9.

Add one paragraph on Implications for policy and practices

How will this finding be useful in other emergencies or strengthen routine healthcare using telemedicine? 

Reviewer 2 Report

Dear authors,  

I think that the topic you have addressed in your paper is very interesting and responds to the current needs in the field of e-heath. However, some clarifications are needed:  

1. At line 16: "371 GPs in Poland participated in the empirical study " At lines 78-80: “The target population of this study were GPs who carried out remote teleconsultations with primary care patients during the Covid-19 pandemic in Poland. In total, 371 respondents took part in the survey. At lines 315-316 “The data collected were complete, with no missing responses. GPs of primary healthcare institutions filled in a total of 361 questionnaires." Please clarify the number of GPs interviewed and how they were selected to form a representative sample for the study.  

2. Please carefully revise Figure 2 (lines 370-371). In the manuscript form there are some misalignments that make the content difficult to understand (see coefficients in the squares related to Confirmatory Factor Analysis).  

3. For a better understanding of the obtained results, please discuss in more details the importance of AUT in the obtained model and what implications does the fact that hypotheses H3a and H3c are not supported (discuss about SIM and IM in relation to NEC).  

I wish you the best of luck in publishing your work!
